# A Theoretical Framework for Rate-Distortion Limits in Learned Image Compression

## Abstract

We present a novel systematic theoretical framework to analyze the rate-distortion (R-D) limits of learned image compression. While recent neural codecs have achieved remarkable empirical results, their distance from the information-theoretic limit remains unclear. Our work addresses this gap by decomposing the R-D performance loss into three key components: variance estimation, quantization strategy, and context modeling. First, we derive the optimal latent variance as the second moment under a Gaussian assumption, providing a principled alternative to hyperprior-based estimation. Second, we quantify the gap between uniform quantization and the Gaussian test channel derived from the reverse water-filling theorem. Third, we extend our framework to include context modeling, and demonstrate that accurate mean prediction yields substantial entropy reduction. Unlike prior R-D estimators, our method provides a structurally interpretable perspective that aligns with real compression modules and enables fine-grained analysis. Through joint simulation and end-to-end training, we derive a tight and actionable approximation of the theoretical R-D limits, offering new insights into the design of more efficient learned compression systems.

## 1 Introduction

In recent years, with the rapid development of deep learning, neural networks have played a significant role in the field of lossy image compression. End-to-end image compression methods (Ballé et al., 2016b; 2017; Theis et al., 2017) based on deep learning have outperformed traditional hand-crafted codecs. A milestone in this line of work is the variational end-to-end compression framework proposed by Ballé et al. (2018), which combines latent variable modeling with a hyperprior network, achieving rate distortion performance comparable to classical codecs such as JPEG2000 (Christopoulos et al., 2000) and BPG (Bellard, 2015). Subsequently, Minnen et al. (2018) introduced context models by combining autoregressive priors with hierarchical priors in the conditional modeling of latents, further improving the rate-distortion (R-D) efficiency. Since then, learned image compression frameworks (Minnen & Singh, 2020; He et al., 2021; 2022) have followed this autoregressive and hierarchical approach to prior modeling, continuously improving performance through more fine-grained design and better utilization of side information.

Despite the impressive empirical performance of these methods, most existing studies primarily focus on engineering optimizations (e.g., network architecture design, quantization strategies), while lacking systematic theoretical lower bound analyses to evaluate their potential optimality. Rate-distortion theory provides the fundamental lower bound on the minimum average bitrate achievable under a given source distribution and distortion metric (Shannon, 1993). For memoryless Gaussian sources, this bound can be computed using the reverse water-filling algorithm (Cover & Thomas, 2006) in information theory. Applying such a theoretical framework to neural image compression allows for quantitative evaluation of the gap between practical models and theoretical optimality, and offers guidance for the design of future models.

This paper builds upon the variational compression framework proposed by Ballé et al. (2018) (hereafter referred to as *Hyperprior*), and systematically analyzes the gap between its performance and the theoretical R-D limit. This includes quantifying the effects of variance estimation error from the

Hyperprior, as well as the inefficiency of quantization methods. Based on this, we propose a novel systematic theoretical framework for analyzing learned image compression and further extend it to account for context modeling, ultimately deriving the theoretical R-D limit under this generalized setting.

The main contributions of this paper are as follows:

- We introduce a principled theoretical framework based on the Hyperprior to estimate the rate-distortion (R-D) limits of learned image compression, providing analytical tools to study the performance gap and contributing factors between learned neural image compression systems and the theoretical R-D limit.

- We decompose the performance gap into three interpretable components: variance estimation, quantization strategy, and context modeling, which are aligned with practical modern learned image compression systems.

- Through joint simulation and training on real-world datasets, we obtain a tight and actionable approximation to the theoretical R-D limit, enabling diagnostic evaluation of current systems and guiding future architectural improvements.

## 2 RELATED WORK

### 2.1 LEARNABLE PARADIGMS FOR IMAGE COMPRESSION

Recent advances in learned lossy image compression follow the VAE-based (**?**) framework proposed by Ballé et al. (2017), where an input image $x$ is mapped to a latent representation $y$ via an analysis transform $g_a$, quantized to $\hat{y}$, and reconstructed by a synthesis transform $g_s$. The model is trained by minimizing the rate-distortion objective. To enable end-to-end training, the quantization is approximated by adding uniform noise $o \sim \mathcal{U}(-0.5, 0.5)$.

To improve entropy modeling, Ballé et al. (2018) introduced a hyperprior network, which estimates spatially-varying entropy parameters (e.g., variance $\sigma^2$) from an auxiliary latent $\hat{z} = Q(h_a(y))$, modeled by a separate network $h_s$.

Minnen et al. (2018) further enhance compression performance by incorporating context models to capture local spatial dependencies in $\hat{y}$, effectively forming an autoregressive model conditioned on past latents. Subsequent works (Minnen & Singh, 2020; He et al., 2021; 2022; Ballé et al., 2020; Cheng et al., 2020; Jiang et al., 2023) have largely followed this design paradigm, leveraging increasingly sophisticated combinations of hierarchical and autoregressive priors to model latent distributions more accurately and achieve improved compression efficiency.

### 2.2 RATE-DISTORTION ESTIMATION

Rate-distortion theory provides a principled foundation for characterizing the minimum rate $R(D)$ achievable under a distortion constraint $D$. However, computing the rate-distortion function $R(D)$ for natural images is notoriously difficult due to the high-dimensional and continuous nature of real-world data, making the classical Blahut-Arimoto algorithm (Blahut, 1972; Arimoto, 1972) difficult to apply directly. Early efforts like Gibson (2017) derived lower bounds using hand-crafted source models, but these are often constrained by simplifying assumptions. More recent approaches estimate $R(D)$ directly from the data. Huang et al. (2020) introduced an AIS-based framework that approximates the $R(D)$ of deep generative models (e.g., VAE (Kingma & Welling, 2013), GAN (Goodfellow et al., 2014), AAE (Makhzani et al., 2015)), treating compression as a lossy coding problem and variational upper bounds to measure rate; Sandwich Bound (Yang & Mandt, 2022) leverages a variational autoencoder to learn latent representations and conditional distributions, yielding paired upper and lower bounds; the NERD (Lei et al., 2022) adopts an end-to-end strategy that jointly optimizes an "optimizer-decoder" pair to minimize mutual information; and the Wasserstein gradient descent (WGD) method (Yang et al., 2023) reformulates the dual of the rate-distortion problem as a functional optimization task with adversarial regularization.

Despite these advances, most data-driven estimators focus on the global $R(D)$ curve and offer limited insight into which components of neural compression models contribute to inefficiency. In

contrast, we take a structurally grounded approach that decomposes performance loss into three key factors: variance estimation, quantization strategy, and context modeling. This decomposition provides a clearer path to closing the gap between practice and theory.

# 3 THEORETICAL RATE-DISTORTION ANALYSIS

In this section, we aim to derive the theoretical R-D limits of modern learned image compression systems. Instead of relying on empirical entropy coding, we propose a mathematically grounded framework that simulates the optimal performance achievable under idealized assumptions. To this end, we decompose the overall R-D performance into three analytically tractable components: variance modeling, quantization, and context prediction.

## 3.1 OVERVIEW OF THE PROPOSED THEORETICAL R-D SIMULATION FRAMEWORK

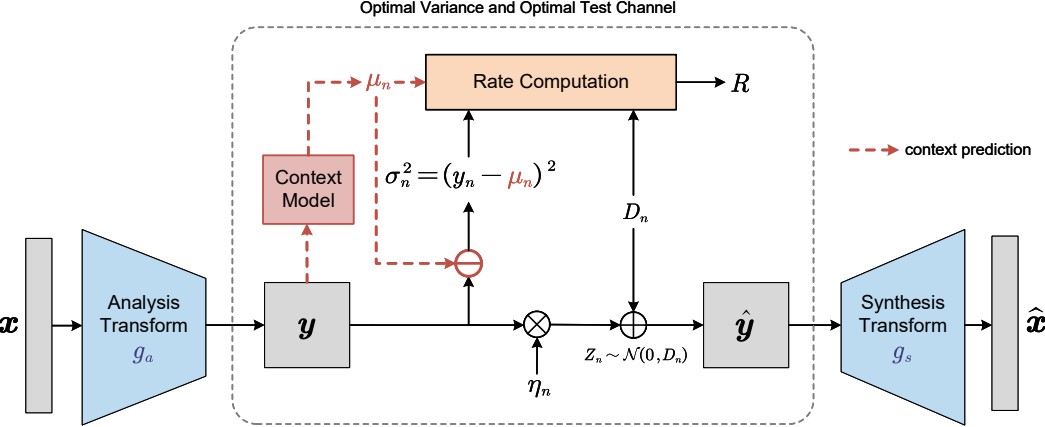

Figure 1: Architecture of the Proposed Rate-Distortion Simulation Framework

Our proposed framework aims to emulate the best-case performance of VAE-based compression models under idealized modeling assumptions. It is built around three core components:

- **Optimal variance modeling:** We replace the hyperprior-estimated variances with optimal variances to match the true distribution.
- **Gaussian quantization:** We substitute the standard uniform scalar quantizer with a continuous Gaussian test channel, aligning with the assumptions of rate-distortion theory.
- **Context modeling:** We incorporate autoregressive mean prediction to capture local dependencies and reduce conditional entropy.

In the following subsections, we analyze each component independently and then combine them into a joint simulation of the theoretical R-D limit.

## 3.2 OPTIMAL VARIANCE MODELING

In the Hyperprior framework, each latent variable is modeled as a zero-mean Gaussian, i.e., $y_n \sim \mathcal{N}(0, \sigma_n^2)$, where the variance $\sigma_n^2$ is predicted by a hyperprior network. This predicted variance is used to construct an entropy model for arithmetic coding.

From an information-theoretic perspective, the expected number of bits required to encode a symbol is given by the cross-entropy between the true and estimated distributions. Therefore, the accuracy of variance estimation directly affects coding efficiency (see Appendix A.1).

Instead of relying on potentially biased hyperprior estimates, we consider an alternative: using the second moment of each latent as its variance. This choice can be interpreted as the result of a maximum likelihood estimation (MLE) procedure under a Gaussian assumption. The following theorem justifies this approach from both an information-theoretic and statistical perspective.

**Theorem 3.1.** *Let $y \sim \mathcal{N}(0, \sigma^2)$ be a latent variable encoded under a Gaussian entropy model with zero mean and predicted variance $\hat{\sigma}^2$. The expected code length is minimized when the predicted variance equals the true second moment of $y$, i.e., $\hat{\sigma}^2 = \mathbb{E}[y^2]$.*

*Proof.* The expected code length for a single symbol is

$$\mathbb{E}[R(y)] = \mathbb{E}[-\log P(y)] = \frac{1}{2}\log(2\pi\hat{\sigma}^2) + \frac{\mathbb{E}[y^2]}{2\hat{\sigma}^2}. \tag{1}$$

Differentiating with respect to $\hat{\sigma}^2$ and setting to zero:

$$\frac{\mathrm{d}}{\mathrm{d}\hat{\sigma}^2}\mathbb{E}[R(y)] = \frac{1}{2\hat{\sigma}^2} - \frac{\mathbb{E}[y^2]}{2(\hat{\sigma}^2)^2} = 0. \tag{2}$$

Solving yields $\hat{\sigma}^2 = \mathbb{E}[y^2]$, completing the proof. $\qquad\square$

In practice, only one sample of $y$ is available at encoding time. Thus, the sample energy $y^2$ serves as an unbiased proxy for $\mathbb{E}[y^2]$, consistent with entropy coding where each code length is computed based on its own likelihood. Replacing the hyperprior-estimated variance with this optimal variance reduces code length and provides a theoretically grounded improvement in coding efficiency.

Modeling latents as Gaussian provides an upper bound under the maximum-entropy principle, which guarantees our rate estimates remain conservative for any $\mathbb{E}[y^2]$-constrained distribution. A detailed discussion of the motivation and implications of this assumption is provided in Appendix E.

### 3.3 Optimal Quantization via Gaussian Test Channel

In addition to the suboptimal variance estimation discussed in Section 3.2, the choice of quantization method also contributes significantly to the R-D gap in Hyperprior frameworks. In practice, each latent variable $y_n$ is quantized using uniform scalar quantization, typically with unit step size: $\hat{y}_n = \text{round}(y_n)$, which is often approximated during training by injecting additive uniform noise: $\tilde{y}_n = y_n + U_n, U_n \sim \mathcal{U}(-0.5, 0.5)$. However, to match the theoretical R-D limit for Gaussian sources with MSE distortion (see Appendix A.2), the noise introduced by quantization must follow a Gaussian distribution. The optimal test channel in this case satisfies:

$$y_n = \tilde{y}_n + Z_n, \quad Z_n \sim \mathcal{N}(0, D_n), \tag{3}$$

which clearly differs from the uniform noise assumption. As such, the actual R-D behavior under uniform quantization *cannot achieve the theoretical limit* given by the Gaussian $R(D)$ function.

To further quantify this mismatch, we compare the effective rate achieved under the actual entropy model used in Hyperprior frameworks with the information-theoretic lower bound.

For each latent variable $y_n$, assuming an estimated variance $\sigma_n^2 = y_n^2$, the probability mass within the quantization bin $[y_n - 0.5, y_n + 0.5]$ is computed under the Gaussian model $\mathcal{N}(0, y_n^2)$. The negative logarithm of this probability gives an estimate of the code length:

$$R_{\text{uniform}} = -\log_2 \int_{y_n - 0.5}^{y_n + 0.5} \mathcal{N}(y; 0, y_n^2)\, \mathrm{d}y. \tag{4}$$

To maintain a fair comparison, we fix the distortion to that of uniform quantization noise, i.e., $D = \frac{1}{12}$, and compute the theoretical rate using the Gaussian rate-distortion function:

$$R_{\text{opt}} = \begin{cases} \frac{1}{2}\log_2\left(\frac{y_n^2}{D}\right), & \text{if } y_n^2 > D, \\ 0, & \text{otherwise.} \end{cases} \tag{5}$$

We conduct a simple simulation by encoding $y_n$ using both methods and compare the rates. As illustrated in Figure 2, under uniform quantization, the coding rate is strictly positive for any non-zero variance and increases monotonically with the variance. In contrast, for the optimal Gaussian test channel, symbols with variance below a certain distortion threshold are not encoded ($R = 0$). Only when the variance exceeds this threshold does the rate become positive (see Appendix A.2). The rate discrepancy between the two schemes is most pronounced near the distortion threshold, and gradually diminishes as the variance increases. Appendix B formally proves that this rate gap asymptotically converges to 0.254 bits (Zamir, 2014). For multiple independent Gaussian sources, the distortion allocation naturally follows the *reverse water-filling principle*, as detailed in Appendix A.3.

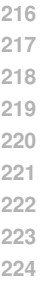
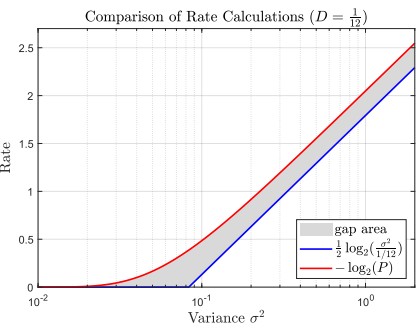
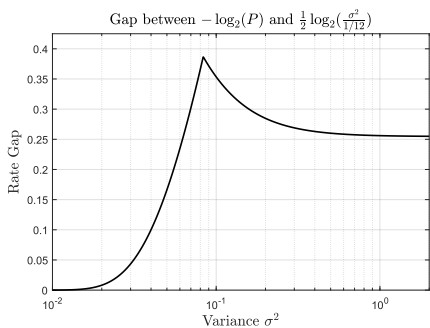

Figure 2: Left (a): Rate-variance curves under uniform quantization and the optimal Gaussian test channel. Right (b): Pointwise difference in coding rate between the two strategies, highlighting the inefficiency of uniform quantization.

### 3.4 JOINT SIMULATION OF VARIANCE AND QUANTIZATION UNDER IDEAL CONDITIONS

To establish a practical framework for the performance ceiling of learned image compression models, we construct a joint simulation that integrates the two previously derived components: optimal variance estimation and Gaussian test channel quantization. This simulation assumes ideal entropy modeling and bypasses actual bitstream generation by analytically computing expected rates and distortions.

Specifically, we assume each latent variable $y_n$ follows an independent Gaussian distribution with optimal variance $y_n^2$. Quantization is modeled using the optimal Gaussian test channel, and distortion is allocated via the reverse water-filling algorithm to minimize total rate under a global distortion constraint.

To simulate the test channel $y_n = \hat{y}_n + n_n$ with $n_n \sim \mathcal{N}(0, D_n)$, we avoid injecting noise directly into $y_n$, which would incorrectly inflate the mutual information. Instead, we first apply a scaling factor and then add noise, yielding:

$$\hat{y}_n = \eta_n \cdot y_n + z_n, \quad z_n \sim \mathcal{N}(0, D_n). \tag{6}$$

The scaling coefficient $\eta_n$ is chosen such that the mutual information between $y_n$ and $\hat{y}_n$ equals the theoretical rate under the optimal Gaussian test channel. This is formalized below.

**Theorem 3.2.** *Let $y_n \sim \mathcal{N}(0, y_n^2)$ and $z_n \sim \mathcal{N}(0, D_n)$ be independent. If $\hat{y}_n = \eta_n \cdot y_n + z_n$, then the mutual information $I(y_n, \hat{y}_n)$ equals $\frac{1}{2} \log_2 \left( \frac{y_n^2}{D_n} \right)$ if and only if*

$$\eta_n = \sqrt{1 - \frac{D_n}{y_n^2}}. \tag{7}$$

The proof of Theorem 3.2 is provided in Appendix C.

Figure 3 illustrates the latent representation $\hat{y}$ produced by the described simulation approach. We observe that in low-variance regions (typically corresponding to smooth background areas such as sky), the latent variables are suppressed to zero, resulting in zero rate allocation according to the reverse water-filling principle. In contrast, under the uniform quantization scheme, all latent variables are added uniform noise and encoded regardless of their variance.

The perturbed latent $\hat{y}$ is passed to the synthesis decoder to reconstruct the image $\hat{x}$. Training is guided by a standard R-D objective:

$$\mathcal{L} = D(\boldsymbol{x}, \hat{\boldsymbol{x}}) + \lambda R. \tag{8}$$

where the distortion is measured by MSE, and the rate is estimated analytically:

$$R = \frac{1}{N} \sum_n \frac{1}{2} \log_2 \left( \frac{y_n^2}{D_n} \right). \tag{9}$$

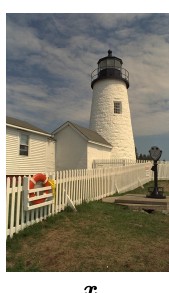 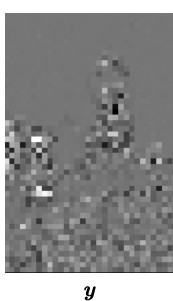 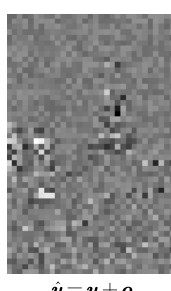 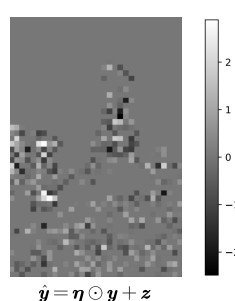

$$x \qquad y \qquad \hat{y} = y + o \qquad \hat{y} = \eta \odot y + z$$

Figure 3: Visualization from left to right: original image, latent variables, uniform quantization output and Gaussian test channel output. Latent variables in low-variance regions (e.g. sky) receive zero rate allocation only under the Gaussian test channel.

### 3.5 INCORPORATING CONTEXT MODELING INTO THE THEORETICAL FRAMEWORK

In the previous analysis, we assumed that the latent variables $\{y_n\}$ are independent Gaussian sources, without considering the correlations (e.g., spatial dependencies in $y$) that may exist between them. However, in practice, even after the analysis transform, local dependencies between latent variables often persist. When correlations exist, the actual mutual information will be strictly lower than the sum of individual rates in Eq. 9 (see Appendix D)

To further improve compression efficiency, modern learned compression frameworks widely adopt *context modeling*, which predicts the local mean of each latent variable using its neighboring latents to reduce the entropy required for encoding. Such context models are typically autoregressive, such as PixelCNN (van den Oord et al., 2016), and effectively capture spatial dependencies in the latent space.

From an information-theoretic perspective, the availability of context reduces uncertainty, as expressed by the following inequality:

$$H(\hat{y}_n \mid \text{context}_n) \leq H(\hat{y}_n). \tag{10}$$

This indicates that a well-designed context model can significantly reduce the conditional entropy of latent variables, and thus lower the average bitrate.

In the idealized compression framework, context modeling is equivalent to introducing a local mean prediction $\hat{\mu}_n$ for each latent. Instead of encoding $y_n$ directly, the system encodes the residual $y_n - \hat{\mu}_n$. The corresponding theoretical encoding rate becomes:

$$R_n = \frac{1}{2} \log_2 \frac{(y_n - \hat{\mu}_n)^2}{D_n}. \tag{11}$$

This formulation naturally integrates the effect of context modeling into R-D estimation: the more accurate the context prediction, the smaller the residual, and hence the lower the number of bits required. It also reveals the functional difference between variance modeling and mean modeling in neural compression systems:

- **Variance modeling** (e.g., via hyperprior networks) focuses on constructing predictive distributions that closely match the true latent distribution, thus reducing the cross-entropy and redundant bitrate.
- **Mean modeling** (e.g., via context networks) lowers the intrinsic entropy of the latent representation by predicting local means that capture spatial dependencies, thereby enabling more efficient compression.

### 3.6 UNIFIED RATE-DISTORTION FORMULATION AND SIMULATION PROCEDURE

To summarize the theoretical framework, we present the final joint formulation of rate and distortion that integrates three key components: optimal variance, optimal Gaussian quantization, and optional

---

**Algorithm 1** End-to-End Simulation of Theoretical R-D Limit

---

**Require:** Input image $\boldsymbol{x}$, analysis transform $g_a$, synthesis transform $g_s$, context predictor $f_\mu$, distortion budget $D$, Lagrange multiplier $\lambda$
**Ensure:** Trained $g_a$, $g_s$ and $f_\mu$ that approximates theoretical R-D behavior
  1: $\boldsymbol{y} \leftarrow g_a(\boldsymbol{x})$                                                    ▷ Encode image to latent
  2: **for all** $n$ in latent indices **do**
  3:      $\hat{\mu}_n \leftarrow f_\mu(\hat{\boldsymbol{y}}_{<n})$                                  ▷ Predict mean using context
  4:      $r_n \leftarrow y_n - \hat{\mu}_n$                                       ▷ Compute residual
  5:      $v_n \leftarrow r_n^2$                                        ▷ Estimate variance of residual
  6:      $\{D_n\} \leftarrow \text{ReverseWaterFilling}\{v_n\}$                  ▷ Bit allocation
  7:      $\bar{y}_n \leftarrow \sqrt{1 - D_n/y_n} \cdot y_n$         ▷ Scale residual to preserve mutual information
  8:      $\hat{y}_n \leftarrow \bar{y}_n + \mathcal{N}(0, D_n)$              ▷ Simulate Gaussian test channel
  9: **end for**
10: $\hat{\boldsymbol{x}} \leftarrow g_s(\hat{\boldsymbol{y}})$                                      ▷ Decode perturbed latents
11: Compute distortion: $D = \text{MSE}(\boldsymbol{x}, \hat{\boldsymbol{x}})$
12: Compute rate estimate:

$$R = \frac{1}{N} \sum_n \frac{1}{2} \log_2 \left( \frac{v_n}{D_n} \right)$$

13: $\mathcal{L} \leftarrow D + \lambda R$
14: Update $g_a$, $g_s$ and $f_\mu$ by minimizing $\mathcal{L}$ via gradient descent

---

context-based mean prediction. This formulation reflects the idealized behavior of a learned image compression system under information-theoretic assumptions. The complete simulation procedure is detailed in Algorithm 1, while the corresponding analytical expressions for rate and distortion are provided in Equation 12. Together, they offer a unified and executable framework for approximating the R-D bound of modern variational compression architectures.

$$\begin{cases} R = \dfrac{1}{N} \sum_{n=1}^{N} \dfrac{1}{2} \log_2 \left[ \dfrac{(g_a(\boldsymbol{x})_n - f_\mu(\hat{\boldsymbol{y}}_{<n}))^2}{D_n} \right] \\[3mm] D = \dfrac{1}{|\boldsymbol{x}|} \sum_{i=1}^{|\boldsymbol{x}|} (x_i - g_s(\hat{\boldsymbol{y}})_i)^2 \end{cases} \tag{12}$$

Here, $g_a(\cdot)$ and $g_s(\cdot)$ denote the analysis and synthesis transforms, $f_\mu(\cdot)$ is the context-based mean predictor, and $D_n$ is the distortion allocated through reverse water filling. The overall end-to-end distortion $D$ is by default measured as MSE, but can also be defined using other differentiable metrics (e.g., MS-SSIM (Wang et al., 2003)), leveraging the expressive capacity of neural networks to learn the corresponding mappings (see Appendix E for further discussion).

## 4 EXPERIMENT

In this section, we conduct a series of experiments to validate the proposed theoretical framework. We compare the estimated variance and uniform quantization with optimal variance and Gaussian test channel quantization to evaluate how much they contribute to the R-D gap. We then jointly simulate optimal variance and quantization to approximate the R-D limit of the Hyperprior framework with context modeling. Finally, we compare our method with existing estimation approaches and previous practical image codecs.

### 4.1 EXPERIMENTAL SETUP

All models are implemented in PyTorch (Paszke et al., 2019). The proposed framework adopts a two-stage training strategy. First, we initialize the model parameters using a pre-trained Hyperprior network. Subsequently, we through an additional $200,000$ training iterations for our framework. This fine-tuning phase employs Adam optimizer (Kingma & Ba, 2014) with a learning rate of $6 \times 10^{-5}$ and a batch size of 100 on a single NVIDIA RTX 4090 GPU. The two-stage training

is adopted purely for efficiency; training the model from scratch yields equivalent performance but requires longer convergence time. We use 100,000 randomly sampled images from the OpenImages dataset (Google Inc.) for training, and evaluate on the Kodak dataset (Kodak, 1993). The Lagrange multiplier $\lambda$ varies from 5 to 700. Compression performance is reported in terms of average bits-per-pixel (bpp), peak signal-to-noise ratio (PSNR, inversely related to MSE) and MS-SSIM. To compare our approach with existing $R(D)$ estimation methods, we additionally conduct experiments on the MNIST dataset (LeCun et al., 1998), which offers a tractable setting for evaluating theoretical R-D bounds. The complete implementation details are provided in Appendix F.

## 4.2 ISOLATING THE IMPACT OF VARIANCE ESTIMATION AND QUANTIZATION

We begin by isolating two key non-idealities in practical compression: inaccurate variance estimation and non-Gaussian quantization noise. To assess the former, we replace the predicted variances from the hyperprior with $\sigma_n^2 = y_n^2$. For quantization, we compare the default uniform scalar quantizer with a simulated Gaussian test channel using fixed distortion $D = \frac{1}{12}$.

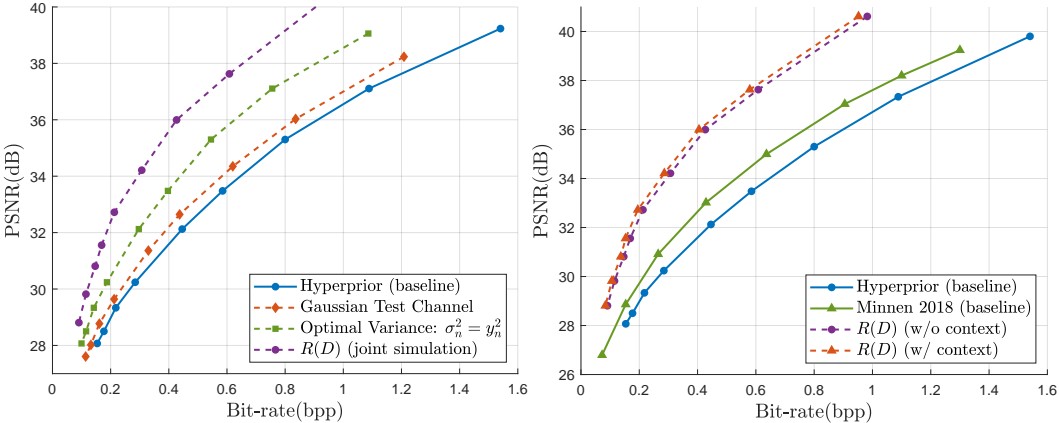

Figure 4: Rate-distortion comparison of different components. Left (a): Comparison between the Hyperprior baseline and variants enhanced with (i) optimal quantization (Gaussian test channel), (ii) optimal variance estimation, and (iii) joint optimization of both. Right (b): Comparison between Hyperprior baseline, the context-enhanced Minnen et al. (2018) model, joint optimization without and with context modeling.

Figure 4(a) presents the $R(D)$ curves comparing the baseline Hyperprior with three progressively refined configurations: (1) optimal variance, (2) quantization via the optimal Gaussian test channel instead of uniform scalar quantization, and (3) a joint simulation combining both components. Each refinement brings the R-D performance closer to the theoretical limit, with the joint configuration achieving the most significant improvement. The results clearly demonstrate that both variance modeling and quantization strategy play critical roles in closing the gap between practical codecs and information-theoretic bounds. These results validate the theoretical insights from Section 3.1 and 3.2, and quantify the individual contributions of each factor to the overall R-D gap.

## 4.3 JOINT SIMULATION RESULTS

Following the procedure described in Algorithm 1, we conduct joint simulation experiments that compute the R-D under context-aware settings. The corresponding results are illustrated in Figure 4(b). The plotted curves reveal a clear performance gap between the baseline Hyperprior model and the theoretical bounds estimated by our framework. Incorporating optimal variance estimation and Gaussian test channel quantization already yields notable gains. Further integrating context-based mean prediction results in an additional bitrate reduction, confirming the importance of capturing spatial dependencies, which indicates that effective context modeling is critical for narrowing the gap between practical codecs and the R-D bound. Similar trends are observed when training with MS-SSIM as the distortion metric. We additionally visualize representative reconstructions from these experiments in Appendix I.

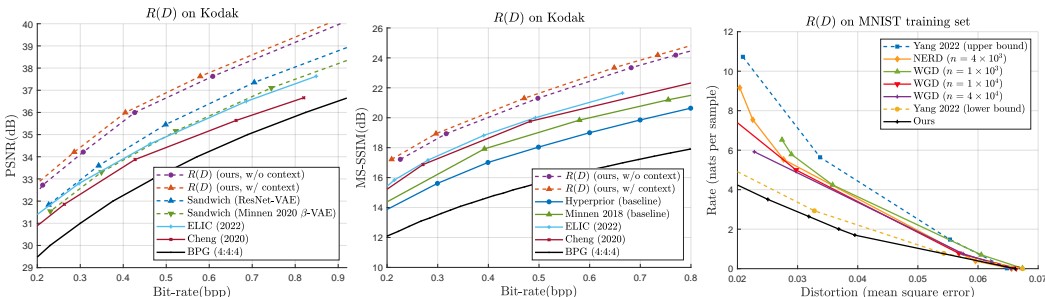

Figure 5: Left (a): Comparison of our theoretical R-D estimates with prior sample-driven estimators and the actual performance of learned image compression models on the Kodak dataset measured by PSNR. Middle (b): Comparison of ours with practical compression models measured by MS-SSIM. Right (c): Comparison of different R-D estimation methods on the MNIST training set.

### 4.4 COMPARISON WITH THEORETICAL RATE-DISTORTION ESTIMATION METHODS

We further compare our proposed method with several representative sample-driven R-D estimation frameworks, including the Sandwich Bound (Yang & Mandt, 2022), NERD (Lei et al., 2022), and WGD (Yang et al., 2023). Figure 5(a) presents the estimated $R(D)$ curves on the Kodak dataset. Our method consistently yields tighter bounds, highlighting its superior accuracy in approximating the information-theoretic limit.

To assess generalization to a different data domain, we also perform $R(D)$ estimation on the MNIST training dataset (LeCun et al., 1998), as shown in Figure 5(c). Notably, due to the small spatial dimensions of MNIST images ($28 \times 28$ pixels), downsampling eliminates nearly all spatial correlations. As a result, the performance gain using context model becomes negligible in this scenario. The results again demonstrate that our approach achieves the closest approximation to the theoretical lower bound among all evaluated methods, confirming the robustness and precision of our estimation framework across datasets.

### 4.5 COMPARISON WITH PRACTICAL LEARNED IMAGE CODECS

Figure 5(a) also shows the empirical R-D curves alongside our theoretical lower bound. While the latest models narrow the gap significantly, our bound remains higher. This suggests that even recent state-of-the-art systems have not yet fully saturated the potential of their architectural priors, especially in high-rate regimes. The proposed framework thus serves as a practical benchmark for future model development.

## 5 DISCUSSION

In this work, we present a theoretical simulation framework based on the Hyperprior architecture to approximate the R-D limit of learned image compression systems, which serves as a fundamental tool for analyzing the performance gap and contributing factors between learned image compression systems and their R-D limits. Specifically, we isolate and quantify the impact of three core components: variance estimation, quantization, and context modeling.

Compared to prior sample-driven estimators (e.g., Sandwich Bound, NERD, WGD), our approach not only offers structural interpretability and actionable guidance, but also consistently achieves lower estimated R-D curves in practice, providing a tighter bound to the theoretical optimum.

While our framework provides a theoretically grounded approximation to the R-D limit, its accuracy still depends on the expressiveness and architecture of the underlying neural networks. Employing more advanced network designs may further tighten the estimated bounds and push the limits of learned compression closer to the information-theoretic optimum.

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

APPENDIX

## A   THEORETICAL FOUNDATIONS OF RATE-DISTORTION ANALYSIS

This appendix provides detailed derivations and theoretical background supporting the main analysis in Section 3.

### A.1   CROSS-ENTROPY AND BITRATE UNDER MISMATCHED DISTRIBUTION

Let $P(y)$ be the true distribution of a symbol and $Q(y)$ be the predicted model used for encoding. According to Shannon's source coding theorem (Shannon, 1948), the average number of bits required to encode a symbol drawn from $P$ using model $Q$ is given by the cross-entropy:

$$H(P, Q) = -\sum_y P(y) \log Q(y) \tag{13}$$

In the case of continuous distributions $p(y)$ and $q(y)$, this generalizes to:

$$H(p, q) = -\int p(y) \log Q(y) \, dy \tag{14}$$

Minimizing this quantity is equivalent to minimizing the KL-divergence between $P$ and $Q$ (or $p$ and $q$) plus the constrained entropy of $P$. Therefore, improving the match between the estimated model and the true distribution reduces the code length.

### A.2   RATE-DISTORTION FUNCTION OF GAUSSIAN SOURCES

For a zero-mean Gaussian source $X \sim \mathcal{N}(0, \sigma^2)$ and squared-error distortion, the rate-distortion function is:

$$R(D) = \begin{cases} \frac{1}{2} \log_2 \left( \frac{\sigma^2}{D} \right), & \text{if } 0 < D < \sigma^2 \\ 0, & \text{if } D \geq \sigma^2 \end{cases} \tag{15}$$

This expression represents the fundamental lower bound on the bitrate required to represent Gaussian sources under mean squared error (MSE) distortion.

**Gaussian Test Channel Interpretation.**   The rate-distortion function $R(D)$ can be interpreted through the lens of a Gaussian test channel:

$$X = \hat{X} + Z, \tag{16}$$

where $Z \sim \mathcal{N}(0, D)$ is independent Gaussian noise with variance equal to the allowed distortion level $D$, and $\hat{X} \sim \mathcal{N}(0, \sigma^2 - D)$. This model ensures that the expected distortion $\mathbb{E}[(X - \hat{X})^2] = D$.

The mutual information between $X$ and $\hat{X}$ in this setting is:

$$I(X; \hat{X}) = \frac{1}{2} \log_2 \left( \frac{\sigma^2}{D} \right), \tag{17}$$

which matches the rate-distortion function $R(D)$. This indicates that the Gaussian test channel achieves the minimum rate required for a given distortion level, making it an optimal solution in the rate-distortion sense.

This interpretation is foundational in information theory and is detailed in standard references such as Cover and Thomas's *Elements of Information Theory* (Cover & Thomas, 2006).

### A.3   REVERSE WATER-FILLING FOR MULTIPLE GAUSSIAN SOURCES

Consider $m$ independent Gaussian sources $X_i \sim \mathcal{N}(0, \sigma_i^2)$ and a total distortion budget $D$. The optimal allocation of distortion $D_i$ to each source minimizes the total rate:

$$R(D) = \min_{\sum D_i \leq D} \sum_{i=1}^{m} \max \left( 0, \frac{1}{2} \log_2 \left( \frac{\sigma_i^2}{D_i} \right) \right). \tag{18}$$

The closed-form solution is found using the reverse water-filling algorithm:

- Choose a water level $\alpha$.
- Set $D_i = \min(\sigma_i^2, \alpha)$ for all $i$.
- Adjust $\alpha$ such that $\sum D_i = D$.

Only sources with $\sigma_i^2 > \alpha$ contribute nonzero rates.

## A.4 Alignment of Latent and Pixel Domain Distortions

We formalize the relation between distortions in the latent and pixel domains, drawing connections to classical transform coding and the optimization process of learned compression models.

**Connection to classical transforms.** In transform coding, orthogonal linear transforms such as the Karhunen–Loève Transform (KLT) preserve mean squared error (MSE) between the original and transformed domains. This property ensures that distortion allocation in the transform domain directly reflects distortion in the pixel domain.

**Role of learned transforms.** In our framework, the analysis and synthesis transforms $g_a(\cdot)$ and $g_s(\cdot)$ are parameterized by neural networks. Although these transforms are nonlinear and not strictly orthogonal, they are trained end-to-end with a pixel-domain distortion objective. This optimization drives the learned transforms to approximate decorrelating mappings that preserve distortion across domains.

**Optimization-driven alignment.** Equation 12 illustrates this principle: the reverse water-filling procedure governs latent-domain allocations $D_n$, while the global distortion $D$ is defined in the pixel space. Because optimization explicitly minimizes $D$, the latent-domain allocations remain aligned with the pixel-domain criterion. This mechanism justifies applying reverse water-filling in the latent space as a tractable approximation to pixel-domain distortion. Moreover, the expressive capacity of neural networks allows the same formulation to be extended beyond MSE, enabling alternative differentiable distortion metrics (e.g., MS-SSIM) to be incorporated within the same framework.

# B Derivation of the Rate Gap Between Uniform Quantization and the Shannon Limit

In high-rate quantization theory, it is known that uniform scalar quantization introduces a fixed gap compared to the theoretical rate-distortion (R-D) lower bound given by Shannon's formula. This section derives the asymptotic rate gap of approximately 0.254 bits per sample, which arises when using a unit-step uniform quantizer with Gaussian sources (Zamir, 2014).

## B.1 Shannon Rate-Distortion Bound for Gaussian Source

For a zero-mean Gaussian source $X \sim \mathcal{N}(0, \sigma^2)$ and mean squared error (MSE) distortion $D$, the Shannon lower bound on the minimum achievable rate is:

$$R_{\text{Shannon}} = \frac{1}{2} \log_2 \left( \frac{\sigma^2}{D} \right). \tag{19}$$

## B.2 Rate of High-Resolution Uniform Scalar Quantizer

Let us consider a unit-step uniform scalar quantizer with interval size $\Delta = 1$. The quantization noise can be modeled as a uniform distribution over $[-\Delta/2, \Delta/2]$. The corresponding quantization distortion is:

$$D_{\text{uniform}} = \frac{\Delta^2}{12} = \frac{1}{12}. \tag{20}$$

The rate required to code a Gaussian source using uniform quantization is given by the asymptotic formula:

$$R_{\text{uniform}} \approx \frac{1}{2} \log_2 \left( 2\pi e \cdot G \right), \tag{21}$$

where $G$ is the normalized second moment of the quantizer, and for uniform quantization, $G = 1/12$.

### B.3 GAP COMPUTATION

The rate gap between uniform quantization and the Shannon bound is therefore:

$$\Delta R = R_{\text{uniform}} - R_{\text{Shannon}} = \frac{1}{2}\log_2(2\pi e \cdot G) = \frac{1}{2}\log_2\left(\frac{\pi e}{6}\right). \tag{22}$$

Substituting values:

$$\Delta R = \frac{1}{2}\log_2\left(\frac{\pi e}{6}\right) \approx 0.254 \text{ bits/sample.} \tag{23}$$

### B.4 INTERPRETATION

This result implies that even under optimal entropy coding, a uniform scalar quantizer incurs an irreducible rate penalty of approximately 0.254 bits/sample compared to the Shannon lower bound. This is due to the shape mismatch between the uniform quantization noise and the optimal Gaussian noise assumed in Shannon's test channel model.

## C PROOF OF THEOREM 3.2 (SCALING FACTOR FOR GAUSSIAN TEST CHANNEL)

**Theorem 3.2** *Let $y_n \sim \mathcal{N}(0, y_n^2)$ and $z_n \sim \mathcal{N}(0, D_n)$ be independent. If $\hat{y}_n = \eta_n \cdot y_n + z_n$, then the mutual information $I(y_n, \hat{y}_n)$ equals $\frac{1}{2}\log_2\left(\frac{y_n^2}{D_n}\right)$ if and only if*

$$\eta_n = \sqrt{1 - \frac{D_n}{y_n^2}}. \tag{24}$$

*Proof.* Since $y_n \sim \mathcal{N}(0, y_n^2)$ and $z_n \sim \mathcal{N}(0, D_n)$ are independent, the linear combination

$$\hat{y}_n = \eta_n \cdot y_n + z_n \tag{25}$$

is also Gaussian with zero mean. Its variance is given by:

$$\text{Var}[\hat{y}_n] = \eta_n^2 \cdot \text{Var}[y_n] + \text{Var}[z_n] = \eta_n^2 y_n^2 + D_n. \tag{26}$$

The mutual information between two jointly Gaussian random variables $y_n$ and $\hat{y}_n$ is:

$$I(y_n, \hat{y}_n) = \frac{1}{2}\log_2\left(\frac{\text{Var}[\hat{y}_n]}{\text{Var}[\hat{y}_n \mid y_n]}\right). \tag{27}$$

Given $\hat{y}_n = \eta_n y_n + z_n$, the conditional variance $\text{Var}[\hat{y}_n \mid y_n]$ is simply the variance of $z_n$, since $z_n$ is independent noise:

$$\text{Var}[\hat{y}_n \mid y_n] = D_n. \tag{28}$$

Thus,

$$I(y_n, \hat{y}_n) = \frac{1}{2}\log_2\left(\frac{\eta_n^2 y_n^2 + D_n}{D_n}\right). \tag{29}$$

To match the desired expression

$$I(y_n, \hat{y}_n) = \frac{1}{2}\log_2\left(\frac{y_n^2}{D_n}\right), \tag{30}$$

we require:

$$\frac{\eta_n^2 y_n^2 + D_n}{D_n} = \frac{y_n^2}{D_n}. \tag{31}$$

Solving for $\eta_n$:

$$\eta_n^2 y_n^2 = y_n^2 - D_n \quad \Rightarrow \quad \eta_n^2 = 1 - \frac{D_n}{y_n^2} \quad \Rightarrow \quad \eta_n = \sqrt{1 - \frac{D_n}{y_n^2}}. \tag{32}$$

This concludes the proof. $\qquad\square$

In addition to the direct derivation, we present an alternative proof that leverages the duality between the rate-distortion function of a Gaussian source and the capacity of a Gaussian channel. This formulation provides further intuition into the role of the scaling factor $\eta_n$ in simulating the optimal Gaussian test channel.

*Proof (from Gaussian test channel perspective).* We consider the classic setting of a Gaussian source and a Gaussian test channel. Let the source $y_n \sim \mathcal{N}(0, \sigma^2)$ and the distortion constraint be $D_n$. According to rate-distortion theory, the minimum achievable rate under mean squared error (MSE) distortion is:

$$R(D_n) = \frac{1}{2} \log_2 \left( \frac{\sigma^2}{D_n} \right), \quad \text{for } D_n \leq \sigma^2. \tag{33}$$

On the other hand, the capacity of a Gaussian channel with signal-to-noise ratio $\text{SNR} = \frac{P}{N}$ is given by:

$$C = \frac{1}{2} \log_2 \left( 1 + \frac{P}{N} \right), \tag{34}$$

where $P$ is the power of the transmitted signal and $N$ is the noise variance.

In our setting, we simulate a test channel of the form:

$$\hat{y}_n = \eta_n y_n + z_n, \quad z_n \sim \mathcal{N}(0, D_n), \quad y_n \sim \mathcal{N}(0, \sigma^2), \tag{35}$$

with $y_n$ and $z_n$ independent.

To ensure that this test channel mimics the optimal rate-distortion behavior, we must force the mutual information between $y_n$ and $\hat{y}_n$ to match the rate-distortion function:

$$I(y_n; \hat{y}_n) = R(D_n) = \frac{1}{2} \log_2 \left( \frac{\sigma^2}{D_n} \right). \tag{36}$$

Now, note that the effective signal-to-noise ratio of the test channel is:

$$\text{SNR}_{\text{eff}} = \frac{\text{Var}[\eta_n y_n]}{\text{Var}[z_n]} = \frac{\eta_n^2 \sigma^2}{D_n}. \tag{37}$$

Thus, the mutual information is:

$$I(y_n; \hat{y}_n) = \frac{1}{2} \log_2 \left( 1 + \frac{\eta_n^2 \sigma^2}{D_n} \right). \tag{38}$$

To make this equal to $R(D_n)$, we equate:

$$\frac{1}{2} \log_2 \left( 1 + \frac{\eta_n^2 \sigma^2}{D_n} \right) = \frac{1}{2} \log_2 \left( \frac{\sigma^2}{D_n} \right). \tag{39}$$

This implies:

$$1 + \frac{\eta_n^2 \sigma^2}{D_n} = \frac{\sigma^2}{D_n} \quad \Rightarrow \quad \frac{\eta_n^2 \sigma^2}{D_n} = \frac{\sigma^2 - D_n}{D_n}. \tag{40}$$

Solving for $\eta_n^2$:

$$\eta_n^2 = \frac{\sigma^2 - D_n}{\sigma^2} = 1 - \frac{D_n}{\sigma^2}. \tag{41}$$

Therefore,

$$\eta_n = \sqrt{1 - \frac{D_n}{\sigma^2}}, \tag{42}$$

which is the desired result.

$\qquad\square$

# D  EFFECT OF SOURCE CORRELATION ON RATE-DISTORTION ESTIMATION

In Section 3, we estimate the theoretical bitrate using the reverse water-filling principle, assuming that latent variables are independent Gaussian sources. However, in practical scenarios, residual correlation may exist between variables, which affects the accuracy of rate estimation. In this appendix, we analyze the impact of such correlation on the total rate-distortion cost.

## D.1  PROBLEM SETUP

Consider two zero-mean Gaussian random variables $X$ and $Y$ with identical variance $\sigma^2$, and a fixed distortion level $D$ for each. The reconstructed variables $\hat{X}$ and $\hat{Y}$ are modeled using an additive Gaussian test channel:

$$\hat{X} = X + Z_X, \quad Z_X \sim \mathcal{N}(0, D), \tag{43}$$

$$\hat{Y} = Y + Z_Y, \quad Z_Y \sim \mathcal{N}(0, D), \tag{44}$$

where $Z_X$ and $Z_Y$ are independent of $X$ and $Y$, respectively. The total rate is given by:

$$R_{\text{total}} = I(X; \hat{X}) + I(Y; \hat{Y}). \tag{45}$$

We now compare $R_{\text{total}}$ under two different assumptions: (1) $X$ and $Y$ are independent, and (2) $X$ and $Y$ are correlated with Pearson correlation coefficient $\rho$.

## D.2  CASE 1: INDEPENDENT VARIABLES

If $X \perp Y$, then the mutual informations decompose as:

$$I(X; \hat{X}) = \frac{1}{2} \log_2 \left( 1 + \frac{\sigma^2}{D} \right), \quad I(Y; \hat{Y}) = \frac{1}{2} \log_2 \left( 1 + \frac{\sigma^2}{D} \right). \tag{46}$$

Hence, the total rate is:

$$R_{\text{ind}} = \log_2 \left( 1 + \frac{\sigma^2}{D} \right). \tag{47}$$

## D.3  CASE 2: CORRELATED VARIABLES

Suppose $X$ and $Y$ have correlation coefficient $\rho$, i.e.,

$$\mathbb{E}[XY] = \rho \sigma^2. \tag{48}$$

The joint distribution of $(X, Y)$ is now correlated, and so are their reconstructions $(\hat{X}, \hat{Y})$.

Let us compute the mutual information between $(X, Y)$ and $(\hat{X}, \hat{Y})$:

$$R_{\text{corr}} = I((X, Y); (\hat{X}, \hat{Y})). \tag{49}$$

Since the system is jointly Gaussian, the mutual information is given by:

$$R_{\text{corr}} = \frac{1}{2} \log_2 \left( \frac{|\Sigma_{\hat{X}, \hat{Y}}|}{|\Sigma_{\hat{X}, \hat{Y}|X,Y}|} \right), \tag{50}$$

where:

$$\Sigma_{X,Y} = \begin{bmatrix} \sigma^2 & \rho\sigma^2 \\ \rho\sigma^2 & \sigma^2 \end{bmatrix}, \quad \Sigma_{\hat{X}, \hat{Y}} = \Sigma_{X,Y} + D \cdot I_2, \tag{51}$$

$$\Sigma_{\hat{X}, \hat{Y}|X,Y} = D \cdot I_2. \tag{52}$$

Then:

$$|\Sigma_{\hat{X}, \hat{Y}}| = \left| \begin{bmatrix} \sigma^2 + D & \rho\sigma^2 \\ \rho\sigma^2 & \sigma^2 + D \end{bmatrix} \right| = (\sigma^2 + D)^2 - \rho^2 \sigma^4, \tag{53}$$

$$|\Sigma_{\hat{X}, \hat{Y}|X,Y}| = D^2. \tag{54}$$

Therefore, the total mutual information is:

$$R_{\text{corr}} = \frac{1}{2} \log_2 \left( \frac{(\sigma^2 + D)^2 - \rho^2 \sigma^4}{D^2} \right). \tag{55}$$

### D.4 COMPARISON AND INTERPRETATION

Comparing the two rates:

$$R_{\text{ind}} = \log_2\left(1 + \frac{\sigma^2}{D}\right), \tag{56}$$

$$R_{\text{corr}} = \frac{1}{2}\log_2\frac{(\sigma^2 + D)^2 - \rho^2\sigma^4}{D^2}. \tag{57}$$

Since $\rho^2\sigma^4 > 0$ for any $\rho \neq 0$, we have:

$$R_{\text{corr}} < R_{\text{ind}}. \tag{58}$$

This shows that treating correlated variables as independent overestimates the total rate. Thus, reverse water-filling performed under the independence assumption yields a conservative upper bound on the achievable bitrate.

## E ON THE GAUSSIAN ASSUMPTION IN THEORETICAL ANALYSIS

In this section, we discuss the Gaussian assumption adopted in our theoretical framework. We first discuss the motivation for modeling latents as Gaussian variables, a common modeling choice adopted by both Hyperprior-based frameworks and our proposed approach. We then analyze how this assumption influences rate estimation in both Hyperprior-based and reverse water-filling formulations. Finally, we provide a maximum-entropy perspective that formalizes the conservativeness of Gaussian-based estimates.

### E.1 WHY MODEL LATENTS AS GAUSSIAN?

Although the true marginal distribution $p^*(y)$ of the latent representation is generally unknown, there are both practical and theoretical motivations for adopting a Gaussian assumption in our framework.

**Analytical tractability.** Our analysis employs mean squared error (MSE) as the distortion metric, which naturally aligns with Gaussian test channels in classical rate-distortion theory. This choice enables closed-form expressions and makes it possible to apply the reverse water-filling theorem. While we primarily focus on MSE for its prevalence and analytical clarity, the framework is not inherently limited to MSE and can in principle be extended to other differentiable metrics.

**Maximum-entropy justification.** From an information-theoretic perspective, the Gaussian distribution maximizes entropy among all distributions with the same variance. Consequently, assuming Gaussian latents leads to conservative rate estimates that serve as valid upper bounds, ensuring the analysis remains meaningful even when the actual latent distribution deviates from Gaussian. We will provide a more detailed discussion later.

**Transform-domain perspective.** Classical signal processing provides additional support for Gaussian modeling. If natural images are approximated as Gaussian mixture models (GMMs), an ideal Karhunen–Loève Transform (KLT) decorrelates the components and maps them into uncorrelated Gaussian variables. However, the ideal KLT is data-dependent and difficult to realize in practice, especially for high-dimensional and non-stationary image distributions. Therefore, we adopt a neural network as a nonlinear and learnable transform to approximate the behavior of the optimal KLT. The latent representation produced by the encoder can be interpreted as a generalized KLT output, which can be reasonably interpreted as approximately Gaussian. Residual dependencies that remain after analysis transforms are explicitly handled by the context model, which captures local correlations through autoregressive mean prediction.

### E.2 RATE ESTIMATION IN THE HYPERPRIOR FRAMEWORK

In the Hyperprior model, each latent variable is encoded using an entropy model $q_\theta(y)$, typically a zero-mean Gaussian with learned variance. The estimated rate corresponds to the expected code length:

$$R_{\text{Hyperprior}} = \mathbb{E}_{p^*(y)}\left[-\log q_\theta(y)\right], \tag{59}$$

which can be decomposed as:

$$R_{\text{Hyperprior}} = H(p^*) + D_{\text{KL}}(p^* \parallel q_\theta), \tag{60}$$

where $H(p^*)$ is the true entropy of the latent variable and $D_{\text{KL}}$ is the Kullback-Leibler divergence between the true distribution and the assumed Gaussian model. This decomposition highlights that Hyperprior-based estimation intrinsically accounts for distribution mismatch via the KL term, yielding a pessimistic (upper-bound) rate.

### E.3 RATE ESTIMATION VIA REVERSE WATER-FILLING

In contrast, our framework directly invokes the analytical rate-distortion function for Gaussian sources:

$$R_{\text{RWF}} = \frac{1}{2} \log_2 \frac{\sigma_y^2}{D}, \tag{61}$$

where $\sigma_y^2$ denotes the second moment of the latent variable, and $D$ is the distortion allocation. This expression assumes that $y \sim \mathcal{N}(0, \sigma_y^2)$, and does not include a divergence term that reflects modeling error. As such, the accuracy of this rate depends on how well the true distribution aligns with the Gaussian assumption.

### E.4 THEORETICAL JUSTIFICATION VIA MAXIMUM ENTROPY PRINCIPLE

According to the maximum entropy theorem (Cover & Thomas, 2006, Theorem 8.6.5), among all distributions with a given variance, the Gaussian distribution achieves the highest differential entropy:

$$H(p^*) \leq H(\mathcal{N}(0, \sigma_y^2)) = \frac{1}{2} \log(2\pi e \sigma_y^2). \tag{62}$$

Therefore, for any $p^*(y)$ with fixed second moment, the corresponding Gaussian-based rate estimation provides an upper bound:

$$R_{\text{true}} \leq \frac{1}{2} \log_2 \frac{\sigma_y^2}{D} = R_{\text{RWF}}. \tag{63}$$

Similarly, in the Hyperprior framework, the presence of $D_{\text{KL}}(p^* \parallel q_\theta)$ ensures that

$$R_{\text{Hyperprior}} \geq H(p^*) \geq R_{\text{true}}. \tag{64}$$

In conclusion, the Gaussian assumption leads to worst-case (maximum entropy) rate estimates, providing a safety margin when the true distribution is unknown. Overestimated rates reduce the risk of underprovisioning in channel or storage resource planning. Both Hyperprior and our methods align with information-theoretic principles, ensuring that rate estimates remain valid even under distributional uncertainty.

## F  MODEL IMPLEMENTATION DETAILS

The detailed architectures of each sub-network in our model are summarized in Table 1. While our theoretical framework is agnostic to specific network designs, we adopt a unified architecture for fair comparison, closely following the configurations used in the Hyperprior (Ballé et al., 2018) model and Minnen et al. (2018).

For experiments conducted on the MNIST dataset, we further introduce a lightweight variant with substantially fewer parameters to prevent overfitting due to the limited data size. The architecture of this compact model is provided in Table 2.

Table 1: Network architecture of the learned compression model and our framework. Each convolution is denoted as "kernel size $k \times k$, $c$ channels, stride $s$".

| Encoder | Decoder | Context Prediction |
|---|---|---|
| Conv: 5×5 c192 s2 | Deconv: 5×5 c$M$ s2 | Masked: 5×5 c$M$ s1 |
| GDN (Ballé et al., 2016a) | IGDN | |
| Conv: 5×5 c192 s2 | Deconv: 5×5 c192 s2 | |
| GDN | IGDN | |
| Conv: 5×5 c192 s2 | Deconv: 5×5 c192 s2 | |
| GDN | IGDN | |
| Conv: 5×5 c$M$ s2 | Deconv: 5×5 c3 s2 | |
| **Hyper Encoder** | **Hyper Decoder** | **Entropy Parameters** |
| Conv: 3×3 c192 s1 | Deconv: 5×5 c192 s2 | Conv: 1×1 c640 s1 |
| Leaky ReLU | Leaky ReLU | Leaky ReLU |
| Conv: 5×5 c192 s2 | Deconv: 5×5 c192 s2 | Conv: 1×1 c512 s1 |
| Leaky ReLU | Leaky ReLU | Leaky ReLU |
| Conv: 5×5 c192 s2 | Deconv: 3×3 c192 s1 | Conv: 1×1 c384 s1 |

Table 2: The lightweight network architecture of our framework trained on MNIST. Each convolution is denoted as "kernel size $k \times k$, $c$ channels, stride $s$".

| Encoder | Decoder |
|---|---|
| Conv: 5×5 c36 s1 | Deconv: 5×5 c36 s2 |
| GDN | IGDN |
| Conv: 5×5 c36 s2 | Deconv: 5×5 c36 s2 |
| GDN | IGDN |
| Conv: 5×5 c36 s2 | Deconv: 5×5 c36 s2 |
| GDN | IGDN |
| Conv: 5×5 c36 s2 | Deconv: 5×5 c1 s1 |

To accommodate different compression regimes, we set the bottleneck channel capacity $M$ based on the target bitrate. Specifically, we use $M{=}192$ for low-rate training scenarios (i.e., bpp $< 0.8$), and increase to $M{=}320$ for high-rate regimes to ensure sufficient representational capacity.

## G  BROADER IMPACTS

This work develops a theoretical framework for analyzing the rate-distortion performance of learned image compression models. While the primary focus is methodological and analytical, the results may inform the design of more efficient compression systems for visual data. Improvements in compression can benefit applications such as image storage, transmission, and on-device processing, particularly in resource-constrained environments.

Our framework does not directly involve human subjects, user data, or social decision-making. As such, we do not foresee immediate negative societal impacts. However, as with any work that optimizes data representation, downstream applications—such as media delivery, surveillance, or generative content—may introduce ethical concerns if used irresponsibly. These broader implications depend on the context of deployment rather than the methodology itself.

The framework may also be extended to other data modalities such as audio, video, or text, which could introduce domain-specific considerations related to perceptual quality, fairness, or privacy. These extensions would require careful adaptation and responsible evaluation.

## H    USE OF LARGE LANGUAGE MODELS

We declare that large language models (LLMs) were employed exclusively to aid in language polishing of this paper. Specifically, the LLM was used for minor improvements in grammar, word choice, sentence fluency and readability. Importantly, no part of the technical content—including the formulation of ideas, theoretical analysis, experimental design, implementation, or interpretation of results—was generated by an LLM. The research contributions, methodology, and conclusions are entirely the work of the authors. The authors take full responsibility for the accuracy and integrity of the paper's content.

## I    QUALITATIVE RESULTS ON KODAK

We visualize five images from the Kodak dataset, each compressed using three different methods: (1) Hyperprior baseline, (2) Joint optimal variance and quantization, and (3) Joint optimization with context prediction. Below each image, we report the corresponding PSNR (in dB) and bitrate (in bits per pixel).

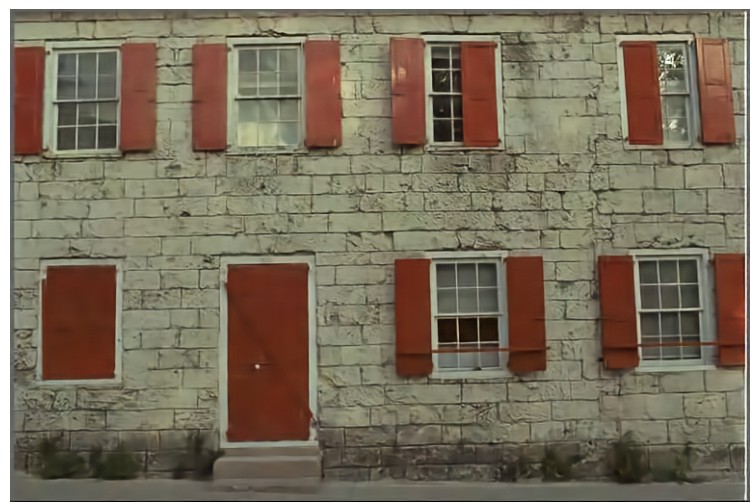

Hyperprior: PSNR = 26.1595, bpp = 0.2675

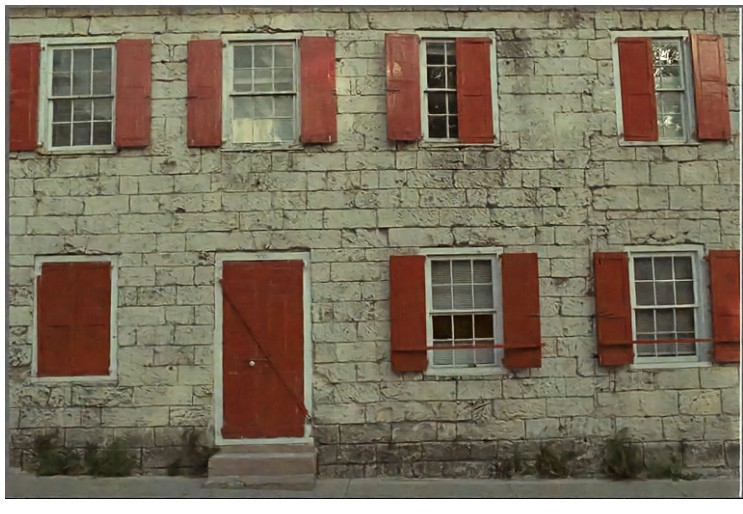

Ours (w/o context): PSNR = 27.7611, bpp = 0.22425

Ours (w/ context): PSNR = 29.0142, bpp = 0.1863

Figure 6: Compression results on kodim_01.png using three methods.

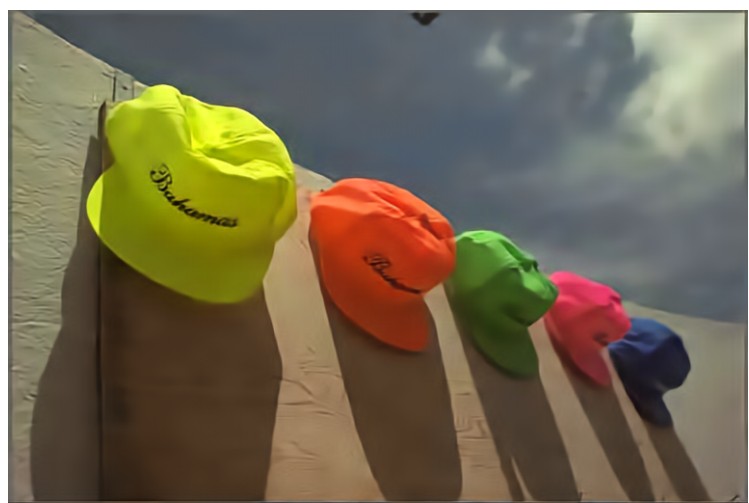

Hyperprior: PSNR = 31.0668, bpp = 0.10352

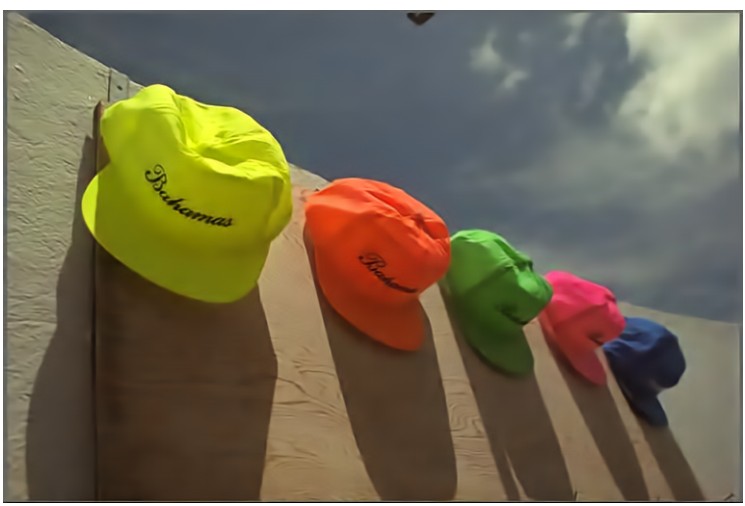

Ours (w/o context): PSNR = 32.6676, bpp = 0.08424

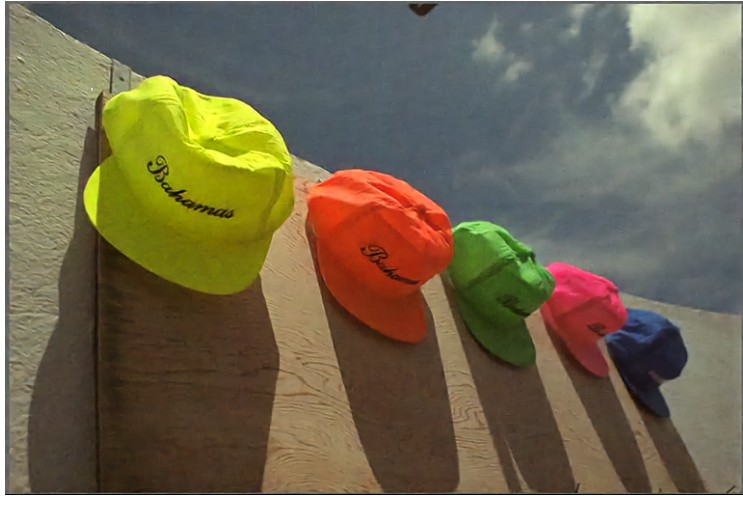

Ours (w/ context): PSNR = 33.9723, bpp = 0.05395

Figure 7: Compression results on kodim_03.png using three methods.

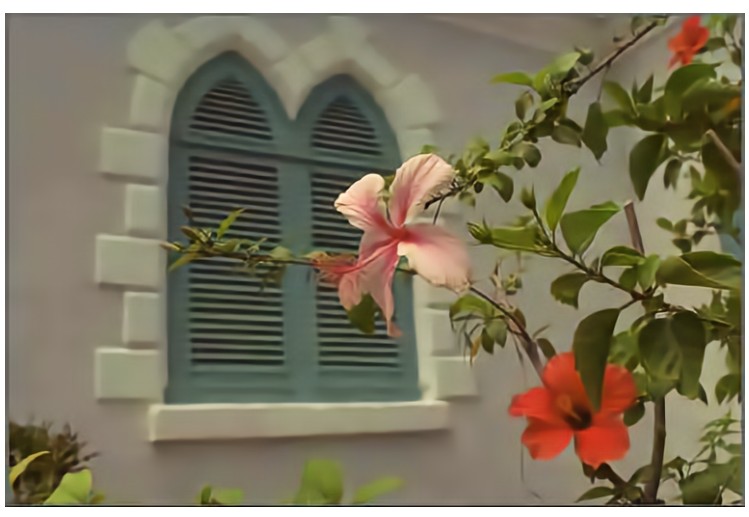

Hyperprior: PSNR = 30.1220, bpp = 0.1587

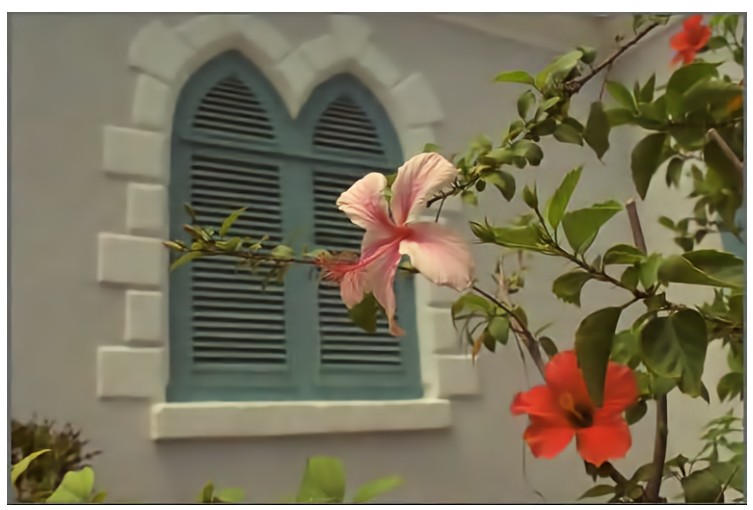

Ours (w/o context): PSNR = 31.9423, bpp = 0.1231

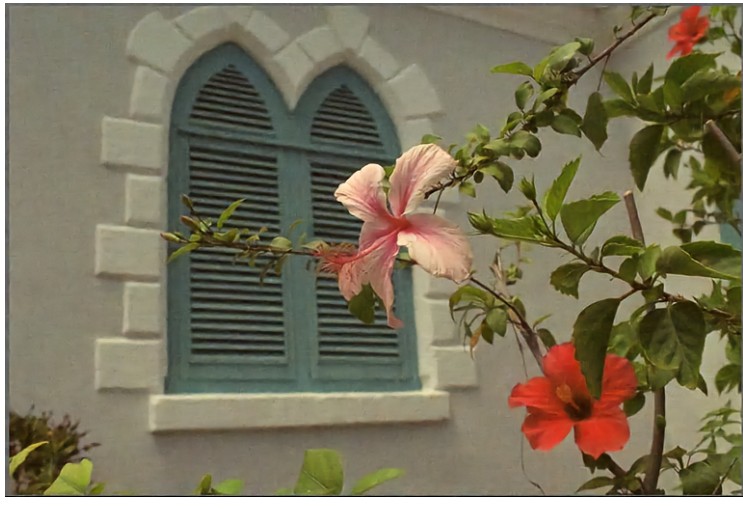

Ours (w/ context): PSNR = 33.2217, bpp = 0.0818

Figure 8: Compression results on kodim_07.png using three methods.

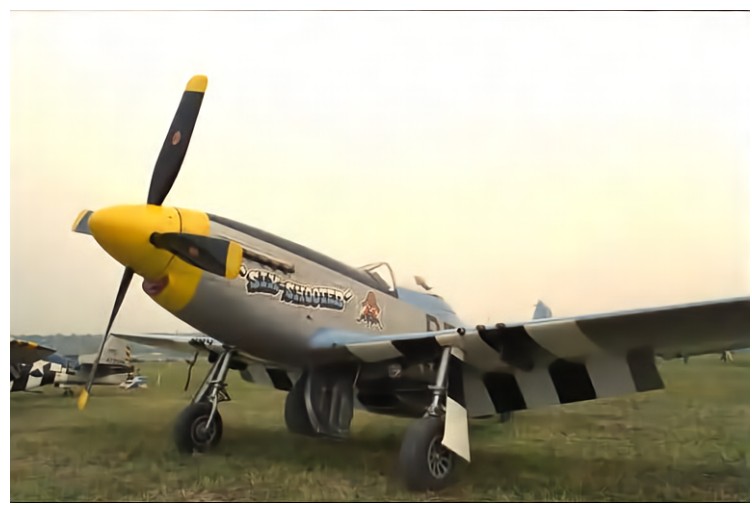

Hyperprior: PSNR = 30.3555, bpp = 0.12093

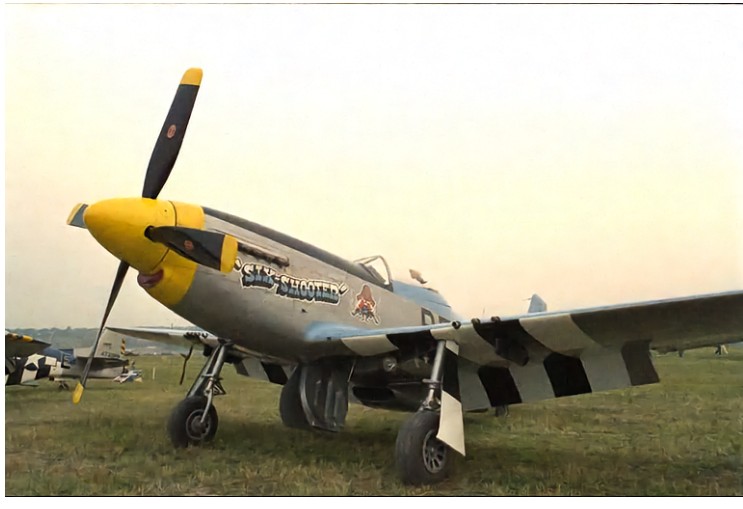

Ours (w/o context): PSNR = 31.8410, bpp = 0.10125

Ours (w/ context): PSNR = 33.1759, bpp = 0.06525

Figure 9: Compression results on kodim_20.png using three methods.

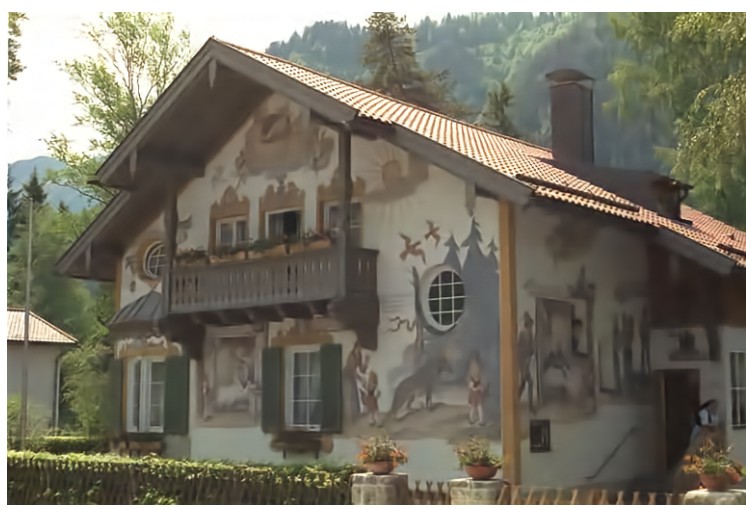

Hyperprior: PSNR = 26.1595, bpp = 0.22485

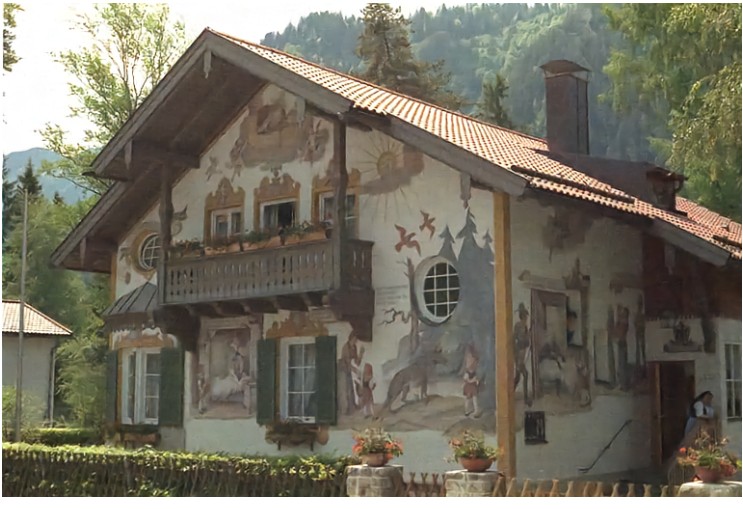

Ours (w/o context): PSNR = 27.8055, bpp = 0.18445

Ours (w/ context): PSNR = 29.3552, bpp = 0.16528

Figure 10: Compression results on kodim_24.png using three methods.

