# OpenReview forum: "A Theoretical Framework for Rate-Distortion Limits in Learned Image Compression"
_ICLR.cc/2026/Conference — ICLR 2026 Conference Withdrawn Submission_

### Official Review · Reviewer_w1ha · 2025-10-24

**Soundness:** 3
**Presentation:** 2
**Contribution:** 3
**Rating:** 4
**Confidence:** 4

**Summary:**

It tries to provide interpretable analysis for learned image compression, and addresses this gap by decomposing the R-D performance loss into three key components: variance estimation, quantization strategy, and context modeling.

**Strengths:**

1. It tries to provide interpretable analysis for learned image compression, and discussses the rate-distortion limits.

2. Motivation is clear and the manuscript is well organized.

3. The analysis of scaling coefficient of $y$ is interesting.

**Weaknesses:**

1. Context model not only predicts the mean value of $y$, but also the variance of $y$. This is not accurate for the whole section 3.5, especially for the equation 11.

2. It is just similar with the prior work "Rethinking Learned Image Compression: Context is All You Need"(https://arxiv.org/abs/2407.11590), which also analyzes the quantization and context model.

3. The importance of $y$ is also dependent on the content of region of image itself, thus the scaling coefficient of $y$ may also not be the best choice. Though I understand it only considers the symbol encoding itself.

4. The visualization in Appendix lacks the comparsion with the prior work.

5. Maybe the results of more datasets like CLIC can make readers have more confidence in this paper.

**Questions:**

See Weaknesses, and I would prefer to improve my rating during rebuttal.

---

### Official Review · Reviewer_1b2F · 2025-10-29

**Soundness:** 2
**Presentation:** 3
**Contribution:** 3
**Rating:** 6
**Confidence:** 2

**Summary:**

In this paper, the authors propose a theoretical analytical framework for quantitatively investigating the rate-distortion performance limits of deep learning image compression models. The framework decomposes the sources of compression efficiency from three interpretable dimensions: Variance Modeling, Quantization Strategy, and Context Modeling. By replacing each of these modules with a theoretically optimal form, the authors establish a computable R-D upper bound and verify the gap between the model and the information-theoretic bound on datasets such as Kodak and MNIST.

**Strengths:**

This paper presents a systematic theoretical framework for analyzing the gap between learning image compression models and the information-theoretic rate-distortion limit. Based on Hyperprior structure, the framework decompose the performance gap from three aspects: variance modeling, quantization method and context modeling, and correspond to them through mathematical derivation and experimental verification. The paper is relatively complete in method design and shows the contribution of each module to the overall rate-distortion performance. The authors provide a clear model structure and training configuration in the experimental section, which makes the study more reproducible. This study establishes a connection between theory and practice, and provides a valuable analysis idea for understanding the performance upper bound of neural compression models and its improvement direction.

**Weaknesses:**

1. Although the PSNR and MS-SSIM results reported in the papers are consistent with theoretical trends, the quantification of statistical fluctuations is lacking. Reproducibility and stability are crucial for research with "theoretical limits" as the core topic.
2. In the experiments, the latent variable dimensions M=192 and M=320 are used to correspond to different code rates, which is reasonable in engineering. However, the effect of model capacity on the theoretical rate-distortion gap has not been systematically analyzed.
3. For a work that aims to provide a "general theoretical benchmark", is validation considered on more types of learned compression models to demonstrate the universality of the framework?

**Questions:**

1. In this paper, the mean of the latent variable is predicted by an autoregressive method to reduce the conditional entropy. In the theoretical framework, is it possible to further reduce the gap between the actual model and the theoretical rate-distortion upper bound by extending the parameterized form of the context model to predict both the variance and the mixture parameter?
2. The paper uses a two-stage training (pre-training with Hyperprior and then fine-tuning), and while claiming "equivalent training results from scratch", it does not provide specific curve or variance validation.
3. This framework assumes that the latent variables follow a uniform Gaussian distribution and does not explicitly consider the statistical differences caused by different semantic contents. Given that compression efficiency in real images often varies by content type (such as texture or smooth regions), do the authors think that the relationship between statistical features of latent variables and semantic information can be further explored in theoretical analysis to explain the differences in compression efficiency at the semantic level?
4. The experiments are based on Kodak and OpenImages. Would the theoretical upper bound fit consistently if tested on more complex or high-resolution datasets?

---

### Official Review · Reviewer_aqSw · 2025-11-01

**Soundness:** 2
**Presentation:** 2
**Contribution:** 2
**Rating:** 4
**Confidence:** 3

**Summary:**

This paper proposes a new theoretical framework to analyze the rate-distortion (R-D) limits of learned image compression, addressing the gap between empirical performance and information-theoretic limits. By decomposing R-D performance loss into key components, the authors provide a deeper understanding of the factors influencing compression efficiency. They introduce a principled method to estimate latent variance based on Gaussian assumptions, offering a more reliable alternative to hyperprior-based methods. Additionally, the framework quantifies the impact of uniform quantization compared to the ideal Gaussian test channel, derived from the reverse water-filling theorem. Context modeling is also incorporated, revealing that accurate mean prediction can significantly reduce entropy. Unlike previous methods, this approach offers a structurally interpretable perspective aligned with real compression systems, enabling fine-grained analysis. Through simulations and end-to-end training, the study delivers a practical and precise approximation of theoretical R-D limits, providing valuable insights for designing more efficient neural compression models.

**Strengths:**

- This paper is theoretically grounded with an information-theoretic perspective.

- Analyzing the R-D limits of existing learned image compression models is interesting.

- The authors motivate general entropy coding frameworks, exhibiting potential impacts.

**Weaknesses:**

- The authors appear to simply believe that applying the reverse water-filling for distortion allocation in the quantization process from y to y_hat would lead to the optimal rate. However, for practical transform coding models like hyperprior, distortion is actually calculated in the pixel domain between $x$ and $\hat{x}$ after going through both analysis and synthesis transforms. In this case, the vanilla reverse water-filling does not necessarily hold. This paper has completely overlooked this issue.

- Contradictory Memoryless Assumption. The paper begins with the memoryless (i.i.d.) assumption, directly applying the reverse water-filling results. However, starting from Section 3.5, the authors reintroduce dependencies between different y variables through "context modeling". At this point, continuing to use memoryless reverse water-filling becomes somewhat self-contradictory. In fact, to my knowledge, the memory version of reverse water-filling does exist.

- The usage of reverse water-filling is questionable and not well-motivated. The authors apply reverse water-filling to allocate distortion $y$ from  to  $\hat{y}$ , but distortion is measured between  $x$ and $\hat{x}$ in practical models. This mismatch is still very vague to me. Although the authors claim that they will introduce more details of reverse water-filling, the current status is not ready for acceptance from my side.

- The practical application of the proposed theoretical framework is questionable compared with the commonly used BD-rate and R-D curve. Although I agree that the previous method (e.g., BD-rate and R-D curve) may be a dataset-driven manner, the proposed method only relies on the model architecture. This arises from the issue that the current method is difficult to generalize to more sophisticated image compression architectures beyond the hypeprior family (e.g., diffusion model or large language model-based compression models in the future).

**Questions:**

Please see the weakness

---

### Official Review · Reviewer_HEhk · 2025-11-01

**Soundness:** 2
**Presentation:** 3
**Contribution:** 2
**Rating:** 2
**Confidence:** 5

**Summary:**

This paper introduces a novel theoretical framework for estimating the rate-distortion (R-D) limits of learned image compression, based on the Hyperprior architecture. It decomposes the performance gap between practical neural codecs and information-theoretic optima into three interpretable components, inlcuding variance estimation (replaced with optimal second-moment estimates), quantization strategy (replacing uniform quantization with Gaussian test channel via reverse water-filling), and context modeling (incorporating autoregressive mean prediction to reduce entropy). The framework simulates idealized behavior through end-to-end training, providing a tight, actionable approximation of the R-D bound. Experiments on Kodak (evaluation), OpenImages (training), and MNIST (comparison with Sandwich Bound, NERD, WGD, etc.) show that this method yields tighter bounds and reveals gaps in SOTA codecs.

**Strengths:**

- It is highly commendable that the authors seek to establish a rigorous theoretical framework for quantifying the performance gap between state-of-the-art codecs and information-theoretic limits.

- The presentation is clear, well-structured, and easy to follow.

**Weaknesses:**

While the paper presents an ambitious and potentially valuable contribution, I have serious concerns about two core aspects of the theoretical framework and its implementation. If unaddressed, these issues would undermine the validity of the analysis, experiments, and conclusions.

- *Scope of the distortion measure in the theoretical analysis.* The framework analyzes optimality in the latent space, with distortion measured between the continuous latent representation and its quantized version. However, in learned image compression, the ultimate distortion of interest is in the image space—i.e., between the original and reconstructed pixels. The analysis appears to overlook the impact of the nonlinear encoder and decoder transforms. The claimed correspondence between latent-space and image-space distortion minimization holds only under restrictive conditions (e.g., if the transforms are distortion-preserving in a specific sense). Without explicit justification or bounds on this approximation error, the theoretical results may not faithfully reflect the true R-D limit of the full codec.

- *Implementation of the optimal variance estimate.* Theorem 3.1 correctly identifies the optimal variance as the second moment of $y$, i.e., $σ^2=E[y^2]$. However, the provided end-to-end simulation (Lines 332 and 391) appears to set $σ^2=y^2$, removing the expectation and treating each symbol as deterministic. This eliminates modeling uncertainty at the decoder, which violates the causal structure of the codec and contradicts the information-theoretic setup.

Additionally, a minor presentational issue: the distinction between the “w/ context” and “w/o context” models in the right panel of Figure 4 is not immediately clear; a brief caption or legend would greatly improve readability.

**Questions:**

I kindly ask the authors to clarify the two major concerns above and, if possible, provide:

- A formal justification (or bounds) showing under what conditions the latent-space distortion analysis accurately approximates the image-space R-D optimum, accounting for the nonlinear transforms.

- Confirmation of the variance implementation and, if $\sigma^2=y^2$ is indeed used, an explanation of how this remains consistent with the causal decoding process and the theoretical optimum.

Addressing these points would significantly strengthen confidence in the framework’s correctness and practical relevance.

---

### Note · Authors · 2026-01-14

**Comment:**

We would like to withdraw this submission. Thank you for your time and consideration.

**Withdrawal Confirmation:**

I have read and agree with the venue's withdrawal policy on behalf of myself and my co-authors.